# Low viscosity of the Earth's inner core

Anatoly B. Belonoshko[1], Jie Fu [2], Taras Bryk [3], Sergei I. Simak [4] & Maurizio Mattesini [5,6]

The Earth's solid inner core is a highly attenuating medium. It consists mainly of iron. The high attenuation of sound wave propagation in the inner core is at odds with the widely accepted paradigm of hexagonal close-packed phase stability under inner core conditions, because sound waves propagate through the hexagonal iron without energy dissipation. Here we show by first-principles molecular dynamics that the body-centered cubic phase of iron, recently demonstrated to be thermodynamically stable under the inner core conditions, is considerably less elastic than the hexagonal phase. Being a crystalline phase, the body-centered cubic phase of iron possesses the viscosity close to that of a liquid iron. The high attenuation of sound in the inner core is due to the unique diffusion characteristic of the body-centered cubic phase. The low viscosity of iron in the inner core enables the convection and resolves a number of controversies.

[1] Department of Physics, AlbaNova University Center, Royal Institute of Technology (KTH), 106 91 Stockholm, Sweden. [2] Faculty of Science, Department of Physics, Ningbo University, 315211 Ningbo, China. [3] Institute for Condensed Matter Physics, National Academy of Sciences of Ukraine, Lviv 79011, Ukraine. [4] Department of Physics, Chemistry and Biology (IFM), Linköping University, SE-58183 Linköping, Sweden. [5] Department of Earth's Physics and Astrophysics, Complutense University of Madrid, E-28040 Madrid, Spain. [6] Instituto de Geociencias (UCM-CSIC), Facultad de Ciencias Físicas, Plaza de Ciencias 1, 28040 Madrid, Spain. Correspondence and requests for materials should be addressed to A.B.B. (email: anatoly@kth.se)

The Earth solid inner and liquid outer cores consist mainly of iron[1]. This was predicted by Birch before the equation of state[2] of iron at high pressure and temperature appeared in the literature. The pressure range of the inner core (IC) is from 330 to 365 GPa[3]. The temperature is known less precisely, since the melting temperatures of iron have not been measured at the IC pressures. Recent assessment[4] suggests that the melting temperature of iron in the center of the Earth could be close to 7000 K. It has long been thought that the stable phase of iron under the IC conditions is the hexagonal close-packed (hcp). However, recent theoretical[4] and experimental[4–6] studies point to the stability of the body-centered cubic (bcc) phase.

There are several properties of the IC that are not compatible with the hcp stability paradigm. One of these is the attenuation of the seismic waves passing through the IC. These seismic waves exhibit strong attenuation[7–9]. That means the seismic waves propagating through the IC are losing their energy due to anelasticity. The attenuation in the IC exhibit a complex behavior but generally is on the level of 0.01–0.001[7,8]. The hcp phase is quite elastic and the quality factor of the hcp phase is on the order of dozens of thousands while the quality factor of the IC is two orders less than that of the hcp phase[10–13].

There were attempts to marry the elasticity of the hcp phase and anelasticity of the IC. Thus, it was suggested[9] that the IC might contain liquid inclusions. About 10% of liquid in the IC would explain the high attenuation[9], however it was shown that the liquid, if once present in the core would have been squeezed out of the IC[13]. Another, more recently suggested explanation would require an anisotropic hcp structure and pairing of impurities[10]. Both of these features do not exist at the high temperature in the IC—the hcp phase becomes isotropic at high temperature[14,15] and the pairing of impurities is improbable due to high entropy. Considering that recent studies[4–6] point to the instability of the hcp phase and stability of the bcc phase in the Core, one needs to estimate the viscosity of the bcc phase under the conditions of the IC. We note, that the viscosity of the solid iron at the pressure 1 bar and high temperature is estimated to be about $10^{13}$ Pa s[15]. The viscosity magnitude of solid iron, if extrapolated to the IC pressure and temperature conditions is expected to be even higher due to the higher pressure.

In contrast with these estimates, we present here the computed viscosity of iron at IC conditions that is extraordinary low for a solid. The viscosity of the bcc iron at the pressure of 365 GPa (pressure at the center of the Earth) is predicted to be in the range $10^{-1}$–$10^2$ Pa s, close to that of a viscous liquid. Such a low viscosity enables convection in the IC, explains experimental controversies and provides explanation for the high attenuation and low shear resistance in the IC.

## Results

**Calculations of mean square displacement and diffusion**. To compute the viscosity of the iron bcc phase we performed ab initio (AIMD) and classical molecular dynamics simulations (see Methods) of the bcc and hcp iron phases. For the AIMD simulations we have chosen $P = 360$ GPa and $T = 7000$ K, the conditions that are close to those at the center of the Earth. We discovered that while atoms in the hcp phase vibrate around their crystallographic positions, the atoms in the bcc phase diffuse along (110) crystallographic planes preserving the bcc structure. We computed the mean square displacement of iron atoms (MSD) both in the hcp and bcc phases (Fig. 1).

$$\text{MSD} = \langle [\mathbf{r}(t) - \mathbf{r}(0)]^2 \rangle \quad (1)$$

where $\mathbf{r}(t)$ is the vector of the atoms configuration at the time $t$. The MSD in both phases behaves in the so called ballistic way in

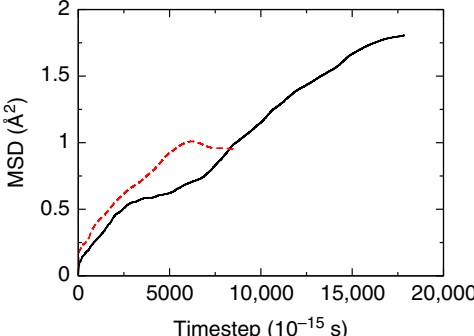

**Fig. 1** Mean square displacement of atoms in two iron phases. Mean square displacement (MSD) computed using configurations obtained by ab initio molecular dynamics at the pressure of 365 GPa and temperature 7000 K. The MSD in the hexagonal close-packed phase (dashed curve) saturates at about 5000 timesteps indicating no diffusion, while the MSD in the body-centered cubic phase (solid curve) steadily grows during the whole simulation indicating the diffusion

the beginning of simulations when atoms are just moving out of their positions increasing the MSD. However, after about 5000 timesteps (the timestep is equal to $10^{-15}$ s) from the start of the simulation the behavior diverges dramatically. The MSD in the hcp phase flattens out indicating the zero diffusivity (see also Supplementary Fig. 1), since the diffusion coefficient $D$ and the MSD are related as follows:

$$6Dt + \text{const} = \text{MSD} \quad (2)$$

in long time ($t$) diffusive regime.

Quite differently, the MSD in the bcc phase steadily increases during the whole simulation (see also Supplementary Fig. 1). The coefficient of diffusion can be calculated and compared to the diffusion coefficient in the liquid iron in the outer core, computed at the same level of theory[16] (that, in fact, is quite similar to the calculations using the embedded atom method[17]). Table 1 provides a summary of diffusion and viscosity of liquid and solid phases of iron under conditions of the IC.

**Stokes–Einstein viscosity calculations**. The viscosity $\eta$ is calculated using the Stokes–Einstein relation[18]

$$\eta = k_B T / 2\pi a D \quad (3)$$

where $k_B$ is the Boltzmann constant and $a$ is the effective atomic diameter, approximately equal to the position of the first peak of the radial distribution function (RDF, see Figs. 2 and 3). Another standard approach for calculations of viscosity of liquids is through the Green–Kubo relation[19,20]. The Stokes–Einstein and Green–Kubo methods provide very close results that can be easily tested for, e.g., liquid argon and were demonstrated for iron[16]. However, to our knowledge, the Green–Kubo approach has not been applied to crystals. The Green–Kubo approach is converging very slowly and would require calculation of correlations on even larger time scale than is needed to compute viscosity via the Stokes–Einstein approach.

The application of the Stokes–Einstein approach to crystals with self-diffusion is legitimate, since the diffusion on a large time and space scales is fully Brownian (Fig. 4; see also Supplementary Fig. 2). To test whether the mechanism of the diffusion is indeed the Brownian one, we performed MD simulation with the embedded-atom method potential[21] involving 250,000 atoms for 5,000,000 timesteps at 360 GPa and 7000 K. The calculated MSD is equal to 20.32 Å, that is on average each atom moved about 4.5 Å. One can see that the atom displacements are randomly

**Table 1 The diffusion coefficient $D$ and/or the viscosity $\eta$**

| Material | $T$, K | $P$, GPa | $D$, $10^{-9}$ m$^2$ s$^{-1}$ | $\eta$, Pa s | Ref. |
|---|---|---|---|---|---|
| BCC | 7000 | 360 | 0.183 | 0.4 | This work, ab initio |
| Liquid | 7000 | 375 | 6.0 | 0.01 | Alfé et al.[16] |
| Liquid | 6000 | 360 | 5.0 | 0.015 | Alfé et al.[16] |
| Liquid | 7000 | 264 | 8.7 | | Koći et al.[17] |
| Inner core | 6000 | 360 | | $<10^{16}$ | Buffett[49,50] |
| BCC | 6000 | 360 | 0.0006 | 130 | This work, EAM |
| BCC | 7000 | 360 | 0.008 | 9 | This work, EAM |
| Liquid inclusions in the inner core | 6000 | 360 | | 250 | Singh et al.[9] |

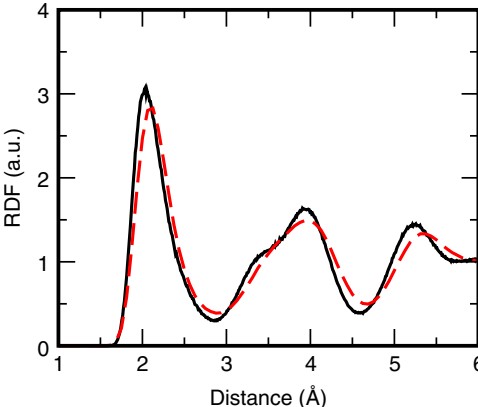

**Fig. 2** Radial distribution functions of the body-centered cubic phase. Two radial distribution functions (RDF) are computed for the system of 1024 Fe atoms using the embedded-atom method (EAM, black solid curve) and the density functional theory (DFT, red dashed curve) at pressure of 360 GPa and temperature of 7000 K (presumably conditions in the center of the inner core). The peaks of the EAM RDF are more distinctive and somewhat displaced in comparison to the DFT RDF. Yet, the similarity is obvious. The heights and positions of peaks of two plots are very close to each other

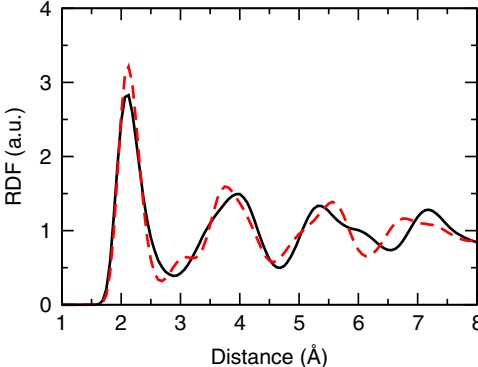

**Fig. 3** Radial distribution functions of two iron phases. The functions are calculated by ab initio molecular dynamics (AIMD) using density functional theory for description of energies and forces. The radial distribution functions are calculated at pressure of 360 GPa and temperature of 7000 K and 1024 Fe atoms. The body-centered cubic phase (black solid curve) radial distribution function (RDF) is less structured than the RDF of the hexagonal close-packed phase (red dashed curve). The peaks of the cubic phase are lower and the minima are higher than those in the hexagonal phase. That can be interpreted as the proximity of the cubic phase to the liquid structure

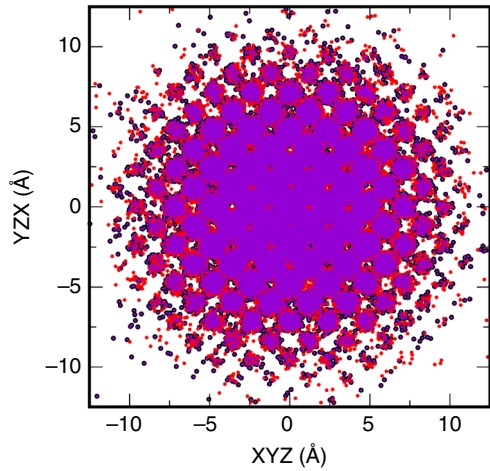

**Fig. 4** Displacement of atoms in the body-centered cubic Fe. The displacements are taken from the final configuration of atoms computed by MD as described in the text. The displacements are shown in three projections. The axes for projections are given as triplets with 1-to-1 correspondence, that is XYZ–YZX corresponds to X–Y, Y–Z, and Z–X projections. The displacements are shown by small circles with correspondingly black, red, and purple colors. The atoms are diffusing through the crystalline positions and at each position they randomly choose one of the six orthogonal directions. Eventually a spherical random distribution of diffused atoms is formed, similar to the diffusion in a liquid state

oriented (Fig. 4) being almost spherically symmetric. Certain modulation due to the crystal structure exists, however, the central part of the cloud of atoms is mostly spherical. The character of the diffusion is quite similar to that in the liquid (see for comparison Supplementary Fig. 2). Clearly, there is no preferable direction for diffusion, the diffusion over long time is completely random and the central part of the cloud of atoms is mostly spherical.

One can see that the viscosity of the bcc iron under conditions of the IC is comparable to the viscosity of the liquid iron, being only 40 times larger (the ab initio calculated) than that in the liquid iron (Table 1). Considering the range of the viscosity that separates the solid hexagonal and liquid phases (about 16 orders of magnitude, see Table 1) the iron in the IC is closer to the liquid than to the solid hexagonal phase viscosity-wise (viscosity of iron at the pressure 1 bar is about $10^{10}$–$10^{13}$ Pa s[22]; experiments[23] at 65 GPa and 2200 K on Fe–Ni alloy being extrapolated to the pressure and temperature in the IC suggest viscosity in the range $10^{20}$–$10^{22}$ Pa s). The size of our ab initio MD simulations is close to the limit of what is currently technically practical. To investigate the impact of the space and time scale increase we performed atomistic MD simulations of iron. At this point one

might wonder why the bcc phase behaves so differently from the hcp phase? The atoms in the bcc phase diffuse and that leads to the efficient mechanism of dissipating the energy of sound waves. The diffusion also explains why RDF of the bcc is rather liquid-like (Fig. 3). The hcp RDF has much sharper and higher peaks than the bcc RDF.

The structure of bcc iron computed for the embedded atom model developed earlier[21] and computed from first principles are rather close to each other (Fig. 4). Therefore, one might expect that the diffusion appears in the case of both models. Indeed, one can see (Table 1) that the atoms diffuse both in EAM and ab initio MD simulations of the bcc phase. The diffusion for the EAM iron is smaller than the diffusion for the ab initio MD iron, yet, in comparison to the latest estimates[23] of the viscosity in the IC equal to $10^{20}$–$10^{22}$ Pa s, we can safely state that the EAM and AIMD lead to the same conclusion: the diffusion of Fe in the IC is high and the viscosity of iron should be lowered by 15–20 orders of magnitude, depending on the temperature in the IC.

**Viscosity temperature dependence**. A more elaborated set of simulations is presented in Fig. 5. We see that the viscosity of iron

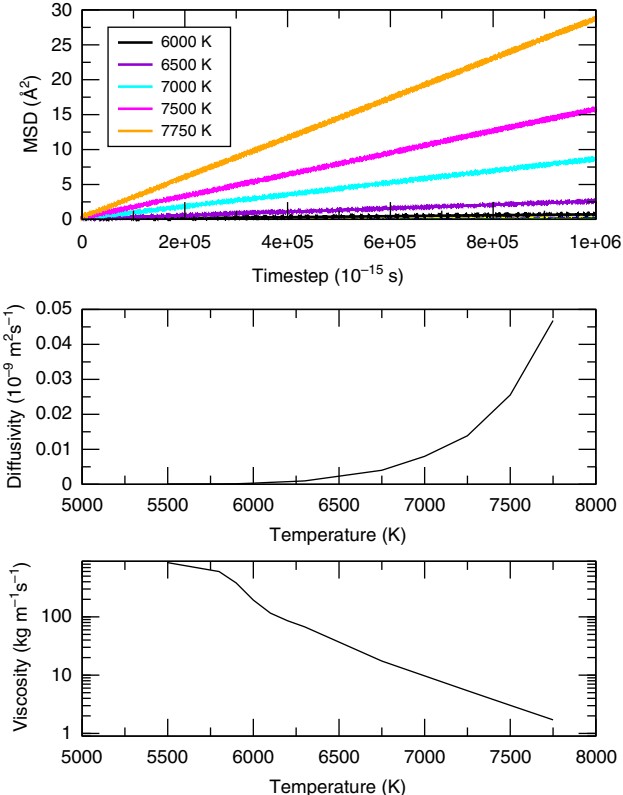

**Fig. 5** Rheology of the body centered cubic iron. Calculated mean square displacement (MSD, upper part), diffusion coefficient (middle), and viscosity (bottom) using the embedded atom method[21] model of iron. Molecular dynamics runs were performed for 2,048,000 atoms for 1,000,000 timesteps at a number of temperatures and the pressure of 360 GPa. The temperature varied from 5500 to 7750 K (see the legend in the upper part). The MSD curves are comparably flat up to 6000 K; above that temperature much stronger diffusion is observed. The diffusion (middle part) becomes very large at temperatures above 6300 K demonstrating exponential dependence on temperature. The viscosity (bottom) behavior is irregular below 6000 K because of comparably low diffusivity but above 6000 K exhibit a regular linear behavior in the logarithmic scale. Even at the lowest temperature of 5500 K the viscosity is still very low

at the pressure of 360 GPa is around 25 Pa s (the one that would maximize the attenuation in the IC[9]) at the temperature between 6500 and 7000 K. This is the temperature which is close to the melting temperature of iron under the pressure in the center of the core. The viscosity grows considerably on temperature decrease and becomes around 130 Pa s at 6000 K (Table 1; this is close to the viscosity of inclusions in Singh et al. paper[9]). Further decrease of temperature leads eventually to very high viscosity.

Considering that the diffusion in the bcc phase is due to the sliding of (110) crystal planes along each other and the fastest direction of sound propagation is ⟨111⟩ it would be interesting to check whether the directions of the attenuation anisotropy and seismic anisotropy cross at the same angles as the corresponding diffusion plane and the fastest sound velocity direction in the bcc structure.

## Discussion

A radical (many orders of magnitude) re-evaluation of the viscosity of the IC opens new avenues for explanation of the processes relevant for the Earth Core. The possibility of the convection in the IC was proposed[24] assuming viscosity $10^9$ Pa s, about 10 orders of magnitude smaller than the latest estimate[23]. As we now know, the viscosity might be much lower than that allowing for convection in the IC. The speed of the convection in the outer core was estimated[25] at the level of $10^{-4}$ m s$^{-1}$. The viscosity in the IC (Table 1) is about 1–4 orders of magnitude larger than in the outer core, depending on the temperature and AIMD/EAM source. Assuming linear dependence of the speed of convection on viscosity, the speed of convection in the IC varies between 0.3 and 300 m year$^{-1}$[26]. Such a motion at the top of the IC can explain the change of the topography at the IC[27] surface. Therefore, the suggested differential rotation of the IC[27] might not be necessary for explanation of the seismic data.

The low viscosity of the bcc phase allows to explain the seismic anisotropy and the low shear modulus[28,29]. Moreover, the bcc phase has the density that matches the density of the IC better than the hcp phase[4]. Indeed, the convection in the IC, enabled by the low bcc viscosity, leads to appearance of texture[24] in the upper part of the IC, which is the most anisotropic part of the IC. The anomalously low shear modulus of the IC is due to the low viscosity of the bcc phase.

The diffusion and liquid-like features of the solid bcc phase allow to resolve the long-standing controversy between the low and high melting curves. In 1993, using DAC with laser heating Boehler reported[30] iron melting curve up to the pressures of almost 2 Mbar. The highest melting temperature at that pressure was under 4000 K. The gradient of the curve was such that extrapolation to the pressures of the IC (330–364 GPa) placed the temperature of the IC under 5000 K. That was in sharp contrast with shock-wave studies. The melting in the early DAC experiments was detected by the visual detection of liquid-like features[30]. As we now know those liquid-like features are the features of the bcc iron. Subsequent theoretical predictions[21,31,32] placed melting curve of iron well above the early DAC measurements, providing melting temperatures of iron at the IC pressures between 6500 and 7500 K, in good agreement with shock-wave data. The controversy seemed to be resolved when Anzellini et al.[33] published the melting curve in good agreement with the theoretical predictions. However, recent experimental study[34] that used X-ray absorption spectroscopy, very sensitive to liquid-like features, confirmed the earlier "low" melting curve. Nowadays we know that the bcc phase does possess liquid-like features (diffusion) that lead to the modifications of the absorption spectra. Therefore, instead of the confirmation of the melting, the new experimental data corroborates the hcp–bcc phase

transition in iron. This is also independently confirmed by the latest experimental results[6].

The proximity of the liquid and bcc iron viscosities is quite remarkable and explains why the idea of liquid inclusions in the IC was so successful. The difference, however, is that one does not need the inclusions—the viscosity of solid iron is close to the viscosity of a liquid. Note, that the scenario where the liquid inclusions are substituted by bcc inclusions in the hcp matrix is not viable because the stable phase of iron in the IC is the bcc phase. The viscosity of the hcp iron is many orders of magnitude larger[23,25] than the viscosity of the bcc phase. The latter is, in fact, quantitatively close to what would be required of the viscosity of the liquid inclusions (at the level of 10% of volume[9]). The quality factor in the IC is in the range of several hundreds[7–13]. To match this quality factor, Singh et al.[9] assumed that the IC contains liquid inclusions, since the hcp—at that time the commonly considered stable phase of iron under the IC conditions—has the quality factor of thousands or even dozens of thousands, being almost perfectly elastic. To match the attenuation in the IC, it was required to have 10% of IC as a liquid with the viscosity of about 250 Pa s. This number is rather close to the range of viscosities computed in this paper for the Fe bcc phase at the IC pressure–temperature conditions (see Table 1). Also, we see (Methods, Supplementary Fig. 3, Supplementary Tables 1 and 2) that the sound damping in bcc iron is similar to the sound damping in liquid. The speed of sound in the center of the core is 11.26 km s$^{-3}$ and this rather well compares to the speed of sound in the bcc phase (12.55 km s$^{-1}$).

The discussion would be incomplete without discussing the impact of impurities on the viscosity of IC. Since iron is denser than the IC, it is likely that Fe in the IC is not pure but rather a mixture of Fe and light impurities, such as Si, S, C, O, and H. While the impurities might enter the Fe lattice in many different ways, if we simply consider a substitution of Fe atoms by light elements without volume change, it would require less than 5% of Fe atoms to be substituted by the impurities to match the density of the IC[3,4]. We note, that all current estimates of the light impurities content in the IC are based on the equation of state of the hcp iron. Rigorous evaluation of the content of impurities, the mechanism of mixing etc. has yet to be measured and computed for the high-PT bcc Fe phase. Currently, it is plausible to suggest that the properties of the 5% impure bcc Fe phase can be rather well approximated by the pure bcc Fe.

Another issue is the amount of Ni in the IC. This is more difficult to estimate, since the atomic weight of Ni is close to that of Fe. Currently, a content of Ni at about 10% is considered as most probable[35]. We also note, that adding Ni to Fe promotes stability of the bcc phase[35]. Since the diffusion and low viscosity of Fe is due to particular structure and dynamic stabilization by high temperature, the addition of Ni will not change the viscosity estimate. Certainly, even though 5% of light elements might lead to a complex mineralogical composition of the IC, it has to be stressed that the major phase is still the almost pure bcc Fe. The impact of Ni and light elements is very likely marginal. The impact of light elements on viscosity of the bcc Fe is best understood if one considers the mechanism of the diffusion in the high-PT bcc Fe. The reason for the diffusion is the dynamic instability of the bcc Fe at high $P$ and low $T$. The (110) layers sliding one along another in combination with relative disorder caused by a high $T$ leads to motion of atoms along these layers. It is quite different from the mechanism of diffusion in dynamically stable phases where the diffusion is controlled by rare jumps over potential barriers. The former mechanism is responsible for the diffusion of Fe atoms in the high-PT bcc structure and it is not sensitive to light elements as long as impurities do not remove the dynamic instability of the high-$P$–low-$T$ bcc Fe phase. Clearly,

under 5% of light elements are not capable of that ab initio MD is routinely performed with 64 atoms—if 3 atoms of these 64 are substituted by carbon that would not change the energy difference between the ideal bcc structure and the wave-mediated one (see Figure 1 in ref. [4]). Considering the limited impact of impurities, we conclude that while quantitatively the viscosity of impure Fe would be somewhat different from the case of pure Fe, qualitatively it is exactly the same.

The stabilization of the bcc phase drastically changes the picture of the IC—from being incompatible with the almost ideally elastic hcp phase it becomes free from multiple enigmas and compatible with the bcc phase. The low viscosity of the bcc phase fits the IC very nicely. Considering the statistical errors of the ab initio MD (because of the small number of atoms and short time of MD runs the errors are comparably high) and deficiencies of the embedded atom model (with negligible statistical errors) we estimate the error of viscosity is mostly due to the differences in model—about an order of magnitude. The impact of statistical errors is much smaller than the order of magnitude. Besides, the viscosity strongly depends on temperature (Fig. 5). Since the calculated viscosity is reasonably close to the viscosity that matches the IC attenuation[7–13] such a strong temperature sensitivity might provide additional constraints on the temperature in the IC. In this case, however, more detailed calculations would be required. Namely, one would need to perform calculations varying the pressure in the IC range (330–364 GPa) and temperature. Note, that the simulations in this paper are performed for pressure 360 GPa. We expect the impact of pressure change within the IC pressure range to be small in comparison to the impact of temperature.

Recent diamond anvil cell studies[36] claim to measure iron at the core conditions. However, the temperature, as suggested by the authors of ref. [37] who analyzed the width of the peaks of the X-ray spectra[36], does not probably exceed 4000–5000 K[37]. This is considerably lower than the IC temperature range. Besides, according to ref. [37], the changes in the X-ray spectra[36] can be explained by the iron carbide formation or the bcc phase stabilization[37].

The most recent experimental study[38] reported melting temperatures of iron up to the pressure of 290 GPa. The study relied on observation of a plateau in the voltage–temperature plot. Such a plateau was observed at low pressure where the melting was independently checked by X-ray diffraction. At higher pressure, where the hcp–bcc transition was predicted and recently experimentally confirmed, the same plateau was observed without independent check by the X-ray. We know now that the bcc phase is similar to liquid diffusion-wise and, therefore, the heat can be efficiently evacuated on the hcp–bcc transition resulting in the plateau similar to that observed on melting.

High-PT studies on iron are full of controversies both on theoretical and experimental side. Quite natural, a general progress in theory and technique suggests preference for recent studies rather than for outdated ones. The recent data supports the stability of the high-PT bcc Fe phase. Considering that latest experiments[5,6] support theoretical predictions on the bcc stabilization on heating under high pressure, there is little doubt that the IC is composed of the bcc phase of iron. Current experiments on properties of iron, such as equation of state, rheology, solubility of light components are all based on the assumption that the stable phase of iron in the IC is the hcp. Taking into account that the hcp and bcc phases have very different properties, it might be that all the extrapolations from the accessible to experiment range of $P$ and $T$ to the $P$ and $T$ of the IC are off by a large amount, similar to the difference in the viscosity of hcp and bcc (i.e., many orders of magnitude). The liquid-like diffusion in a monatomic solid is a unique phenomenon. The fact, that this

phenomenon allows to explain such an enigmatic property of IC as the high attenuation is quite unlikely to be fortuitous. It is timely that the focus of the Core research is concentrated on the bcc phase.

The body-centered phase of iron stable under the pressure and temperature of the Earth IC is the material with uniquely low viscosity. Considering that iron is the major component of the IC, we can conclude that the IC has a very low viscosity, so far it was believed that only liquids can have such viscosity. This material is truly unique and that explains the unique properties of the IC. Such features of the IC as the high attenuation, its differential rotation, the convection, formation of the lattice preferred orientation resulting in the seismic anisotropy, and the low shear resistance receive a natural explanation.

## Methods

**Ab initio molecular dynamics.** The energies and forces were calculated within the framework of the frozen-core all-electron projector augmented wave (PAW) method[39], as implemented in the Vienna Ab initio Simulation Package (VASP)[40–42]. The energy cut-off was set to 400 eV. Exchange and correlation potentials were treated within the generalized gradient approximation (GGA)[43,44]. Fourteen electrons were considered as valence, therefore any overlapping of core states at high pressures and temperatures was avoided. The finite temperatures for the electronic structure and force calculations were implemented within the Fermi–Dirac smearing approach[45]. All necessary convergence tests were performed, the electronic steps converged within 0.0001 meV atom$^{-1}$. The ab initio molecular dynamics runs have been performed in the NPT (N—number of atoms, P—pressure, T—temperature) ensemble for a given pressure and temperature. We used supercells with 1024 atoms for both bcc and hcp runs. Tests have shown that $\Gamma$-point is sufficient at these sizes. The timestep was set to 1 fs and the runs continued for up to 18,000 timesteps.

**Classical molecular dynamics.** Classical MD runs have been performed in NVT and NPT ensembles using the Lennard–Jones potential for Ar and embedded-atom potential for Fe[21]. The number of particles in classical MD simulations ranged from 4000 to 2,048,000 and were run for up to several millions of timesteps. The MD simulations have been performed using LAMMPS package[46].

**Calculations of sound dumping.** For comparison with the diffusion in the bcc phase of Fe we performed a similar simulation of liquid Ar interacting with Lennard–Jones potential (Supplementary Fig. 2). While some differences in the diffusion pattern do exist, the diffusion in solid Fe (Fig. 4) and liquid Ar (Supplementary Fig. 2) is very similar. The central part of the cloud of atoms is mostly spherical in both cases. We note, that on further time increase the similarity will be even more pronounced due to the growth of the central spherical region (Fig. 4). Clearly, there is no preferable direction for diffusion, the diffusion over long time is completely random and the Stokes–Einstein estimate is justified.

We have studied the propagation of sound excitations in condensed matter by molecular dynamics simulations using the interaction models developed earlier, namely the Lennard–Jones and the EAM[21] potentials. A sound wave represents the periodic propagation of densification and rarefaction. Propagating acoustic modes can be described and analyzed by analytical models for corresponding time correlation functions[47]. These time correlation functions can be directly calculated from molecular dynamics simulations. In general, the damping of sound in studied systems can be analyzed from imaginary part of density response function Im $\chi(k, t)$, which is connected via time derivative to the regular density–density time correlation function $F_{nn}(k, t)$[48].

For liquids the long-wavelength asymptotic equation for the density–density time correlation function $F_{nn}(k, t)$ reads[47,48]

$$F_{nn}(k, t)/F_{nn}(k, t = 0) = A_{nn}e^{-D_T k^2 t} + [B_{nn}\cos(c_s kt) + D_{nn}(k)\sin(c_s kt)]e^{-\Gamma k^2 t} \tag{4}$$

where $A_{nn}(k \to 0) = 1 - \gamma^{-1}$, $B_{nn}(k \to 0) = \gamma^{-1}$, $\Gamma = (D_L + (\gamma - 1)D_T)/2$ is the sound damping coefficient expressed via kinematic viscosity $D_L$ and thermal diffusivity $D_T$, $\gamma = C_P/C_V$ is ratio of specific heats, $c_s$—speed of sound and $D_{nn}(k) = \frac{3\Gamma - D_L}{\gamma c_s}k$. Equation 4 is consistent with the continuity equation and hence with the longitudinal current–current correlations

$$F_{JJ}^L(k, t) = -\frac{1}{k^2}\frac{d}{dt^2}F_{nn}(k, t) \tag{5}$$

with the correct initial value (zero sum rule for current autocorrelations) $F_{JJ}^L(k, t = 0) = k_B T/m$, where $T$ and $m$ are temperature and atomic mass, respectively[48]. Each of the three time correlation functions, $F_{nn}(k, t)$, $\chi(k, t)$, $F_{JJ}^L(k, t)$, contains all the information about spatial dispersion and damping of longitudinal acoustic modes. Note, that the relaxation mode in hydrodynamic

Eq. 4 corresponds to the thermal relaxation with the thermal diffusivity coefficient $D_T$ (diffusivity of local temperature), that contributes to sound damping in liquids. In crystals, however, sound damping is defined by anharmonicity of phonons and electron-ion coupling. At high temperatures the crystal structure can coexist with non-zero diffusion of atoms along some symmetric lines[4]. In that case the release of the stress caused by structural relaxation should appear. An account for such a relaxation can be made within the viscoelastic model, for which the density–density time correlation function can be represented as follows:

$$F_{nn}(k, t)/F_{nn}(k, t = 0) = A_{nn}e^{-t/\tau_\sigma} + [(1 - A_{nn})\cos(c_s kt) + D_{nn}(k)\sin(c_s kt)]e^{-\Gamma k^2 t} \tag{6}$$

where $\tau_\sigma$ is the specific timescale of stress relaxation, related to kinematic viscosity. The fits to molecular dynamics data are provided in Supplementary Table 1 (for liquid) and in Supplementary Table 2 (for crystals).

## Data availability
The datasets generated during and/or analyzed during the current study are available from the corresponding author on reasonable request.

## Code availability
The Vienna Ab Initio Simulation Package (VASP) is a proprietary software available for purchase at https://www.vasp.at/. The LAMMPS (Large-scale Atomic/Molecular Massively Parallel Simulator) package is available at https://lammps.sandia.gov/.

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

## Acknowledgements

Computations were performed using the facilities at the Swedish National Infrastructure for Computing (SNIC) located at the National Supercomputing Center in Linköping. The authors also wish to thank the Swedish Research Council (VR) for financial support (Grants 2013-5767, 2014-4750, and 2017-03744) and National Natural Science Foundation of China (Grant No. 11804175). A.B.B. and T.B. acknowledge support from Olle Engkvist Byggmästare Foundation. S.I.S. acknowledges the support from the Swedish Government Strategic Research Area in Materials Science on Functional Materials at Linköping University (Faculty Grant SFO-MatLiU No. 2009 00971). M.M. acknowledges financial support by the Spanish Ministry of Economy and Competitiveness (CGL2013-41860-P and CGL2017-86070-R).

## Author contributions

A.B.B. and M.M. designed the study. A.B.B. performed calculations and wrote the paper. J.F., T.B., and S.I.S. performed calculations. All authors discussed the results and contributed to paper writing.

## Additional information

**Competing interests:** The authors declare no competing interests.

