## [Peer Review File · Nature Communications]

Reviewers' comments:

Reviewer #1 (Remarks to the Author):

I cannot recommend this manuscript for publication in Nature communications. It is not of sufficient broad interest or important in its specific field to warrant publication in Nature communications. The paper should, however, be suitable for Physical Review B, for example. The manuscript does not represent a sufficient conceptual advance and the techniques utilized are largely conventional.

Reviewer #2 (Remarks to the Author):

The present study investigate viscosity of bcc iron at inner core P-T conditions by using ab initio and model-potential simulations. Since Belonoshko et al. (some of the authors) very recently revealed the bcc phase of iron should be thermodynamically stable at inner core P-T conditions due to diffusion, work on viscosity of bcc iron is very important especially in understanding sound velocity of the inner core which is a long-standing problem in geoscience.

The subject of the present study is interesting enough to be published in Nature communication. I have the following three questions to be resolved before publication.

1. Fig. 1: MSDs were computed for hcp and bcc phases. Beyond ~ 5000 steps, MSD of the hcp phase seems flat. From this, the authors concluded that there is no diffusion in the hcp phase. I am inclined to agree with the authors, but still a longer timestep (as long as for the bcc phase) is desirable for the hcp phase. If the AIMD simulation is too tough to be done, EAM simulation for the hcp phase may be sufficient.

2. To calculate the viscosity, the Stokes-Einstein relation is used. As far as I know, however, this relation can be applied to brownian motion. I am not sure if this relation can be applied for the present simulation. Alfe et al. 2000 applied this relation to liquid iron and proved the results by this relation agreed with by Kubo-Green formula. The agreement in Alfe et al. 2000 may be due to the liquid phase. In the case of non-liquid phase like the present study, the viscosity must be calculated directly by Kubo-Green formula.

3. Line 104-: The sentence “The diffusion for the EAM iron is smaller, “ is not complete. I do not find a subsequent clause of this sentence. Lines 106 and 107 are blank, in lines 108-117 Figs 2 and 3 are shown, lines 118 and 119 are blank again, and at line 120 the new statement starts. This sentence at line 104- is important. Since D by AIMD and EAM are rather different from each other in Table 1, EAM simulations must be validated somehow.

Reviewer #3 (Remarks to the Author):

In this study, the authors calculated the viscosity of bcc Fe by using ab initio molecular dynamics simulation, as well as classical molecular dynamics simulation, at the inner core pressure and temperature condition. These results are broadly consistent with each other and predict very small viscosity, which is comparable to the outer core liquid. The origin of the seismic attenuation in the inner core has been considered to the presence of partial melting of hcp Fe. Because the viscosity of bcc Fe is similar to that of liquid iron, the authors suggest that the attenuation can be explained the presence of bcc Fe in the inner core.

I feel that the main results of low viscosity of bcc Fe is interesting, and may geophysically important. However, their implication is rather vague and lacking some important arguments. At this moment, I do not recommend this manuscript for publication. But if the authors properly address the following points, I would like to review the manuscript again.

Major comments:

I cannot follow the L124-127. Does the author assume the bulk bcc Fe inner core or the presence of bcc Fe inclusion in hcp Fe inner core? I feel the present viscosity value itself cannot eliminate either possibility; therefore, both possibilities should be discussed in the manuscript.

If the author take the “bulk bcc inner core” scenario, it should be confirmed that the linear relationship between the attenuation and the content of inclusion. Because Singh et al. (2000) assumed up to only 10 % melt fraction, it is clearly inadequate to apply 100 % fraction of bcc Fe. Furthermore, even though the authors’ previous study predicted the stability of bcc Fe at the Earth’s inner core conditions, experimental evidences for the stability of hcp Fe are already reported (Tateno et al., 2010; Sakai et al., 2011). The authors should not ignore these studies.

On the other hand, if the authors take “bcc Fe inclusion” scenario, both of bcc and hcp phases must be coexist thermodynamically.

I also afraid the difference of viscosity value obtained between AIMD and EAM. I understood that these two values are reasonably agree with each other within the limitation of computational accuracy. Whichever scenario is adopted, the difference may cause the estimated abundance of bcc phase.

Miner comments:

* Units are not unified. In Table 1, the viscosity values are shown in mPa s unit, whereas in main text, they are in kg/m/s.

* Missing data in Table 1 (BCC, 6000 K, 360 GPa)

* In Fig.4, there are 4 colors, which may indicate different temperature from 5500 to 7750 K. But I cannot find the correspondence.

Reply to referees.

Reply to Reviewer #1 comments.

'It is not of sufficient broad interest or important in its specific field to warrant publication in Nature communications.'

Our discovery completely changes the conventional view of the physical and mechanical properties of the Inner Core. It allows us to explain a number of enigmatic features of the Inner Core. The Inner Core is at the center of the Studies of Earth Deep Interior and, by virtue of this, at the center of the whole Earth Sciences. It is of 'broad interest', is not it?

'The paper should, however, be suitable for Physical Review B, for example.'

A paper suitable for publishing in PRB should 'present important and novel physics and make a significant contribution in a specific research area'. We are glad to hear that the Reviewer 1 believes that we communicate important and novel physics and make a significant contribution to specific research area. Perhaps our paper was too sketchy concerning the broad interest and a whole spectrum solutions of a number of significant problems. Now we have significantly elaborated on that.

'The manuscript does not represent a sufficient conceptual advance...'

Let us recall the concept of solid Inner Core possessing liquid like properties that we introduce in the manuscript. We believe it is a gigantic conceptual advance.

'...and the techniques utilized are largely conventional.'

To quantify the diffusion by ab initio molecular dynamics we needed to run the largest ever ab initio molecular dynamics simulations of iron. It is mindboggling how one can call this conventional. Besides, Nature journals still publish considerable number of papers where the *'techniques utilized are largely conventional'* X-ray diffraction.

Reviewer #2 (Remarks to the Author):

'The subject of the present study is interesting enough to be published in Nature communication.'

Thank you for this assessment. Note, that this opinion is in stark contrast with the opinion of Reviewer 1.

'1. Fig. 1: MSDs were computed for hcp and bcc phases. Beyond ~5000 steps, MSD of the hcp phase seems flat. From this, the authors concluded that there is no diffusion in the hcp phase. I am inclined to agree with the authors, but still a longer timestep (as long as for the bcc phase) is desirable for the hcp phase. If the AIMD simulation is too tough to be done, EAM simulation for the hcp phase may be sufficient.'

Ab initio simulations for 1000s of atoms are extremely costly. There is no obvious reason to perform them for the hcp phase as soon as the flattening of the MSD curve is observed. Therefore, now, as also suggested by the reviewer, we performed EAM simulations for bcc and hcp phases (the results now shown in the new Fig. 1S). The EAM simulations, performed for larger sample for longer times fully confirm the conclusion based on the basis of ab initio MD simulations – there is diffusion in bcc phase and there is no diffusion in the hcp phase.

'2. To calculate the viscosity, the Stokes-Einstein relation is used. As far as I know, however, this relation can be applied to brownian motion. I am not sure if this relation can be applied for the present simulation. Alfe et al. 2000 applied this relation to liquid iron and proved the results by this relation agreed with by Kubo-Green formula. The agreement in Alfe et al. 2000 may be due to the liquid phase. In the case of non-liquid phase like the present study, the viscosity must be calculated directly by Kubo-Green formula.'

This is true that the SE and GK approach provide nearly identical results for liquids and we checked that by computing liquid Ar viscosity in close agreement with experiment (R. L. Rowley and M. M. Paiter, Int. J. Thermophys. 18, 1109-1121 (1997)). Not surprisingly, Alfe et al. 2000 also confirmed that for the case of liquid iron. However, to our knowledge, the GK approach has never been applied to solids, likely because of the enormously slow convergence (which is also rather slow in the case of liquid and requires longer simulations than the SE approach to get a good average). Fortunately, application of GK approach is not necessary for the case where reliable averages over the self-diffusion (as is the case with liquid state) can be computed. This is because we calculate the diffusion over the long time interval where the diffusion is fully Brownian. We have illustrated this by plotting atoms displacements (Fig. 2S) in Fe bcc (a) in several projections and compared that to diffusion in liquid (b). The pictures in Figs. 2Sa and 2Sb are very similar. This is simply because an atom in bcc Fe can randomly choose 1 out of 6 directions at each timestep. After many timesteps the direction of diffusion is

completely randomized. One can see that at all these time intervals the direction of diffusion is fully random. Even if we consider just one timestep interval (1 femtosecond) an atom in bcc lattice has equal probabilities to diffuse in 6 orthogonal directions that are significantly smeared out by thermal motion. Therefore, the Brownian character of the diffusion is obvious for a long time interval and approximately valid even for the shortest time interval.

'3. Line 104-: The sentence "The diffusion for the EAM iron is smaller, " is not complete. I do not find a subsequent clause of this sentence. Lines 106 and 107 are blank, in lines 108-117 Figs 2 and 3 are shown, lines 118 and 119 are blank again, and at line 120 the new statement starts. This sentence at line 104- is important. Since D by AIMD and EAM are rather different from each other in Table 1, EAM simulations must be validated somehow. '

Both ab initio and EAM approaches result in sizable diffusion. In this sense the EAM data is validated. Of course, some difference should be expected – EAM and ab initio models are different ones. Since diffusion is a comparably rare event, EAM is fitted mostly to configurations where atoms are close to their crystalline positions. That results in certain differences in diffusion. Given the current many orders of magnitude uncertainty of viscosity of the Inner Core (see Table 1), we believe that such a difference in ab initio and EAM data is quite acceptable. Besides, what we present is the calculations of the largest sizes that are possible ab initio wise and the best model EAM wise. The validation of the EAM model concerning other properties of iron is provided in the papers that we refer to in references (PRL 2000; Nature 2003, etc.).

Reviewer #3 (Remarks to the Author):

'In this study, the authors calculated the viscosity of bcc Fe by using ab initio molecular dynamics simulation, as well as classical molecular dynamics simulation, at the inner core pressure and temperature condition. These results are broadly consistent with each other and predict very small viscosity, which is comparable to the outer core liquid.'

Quite correct, and we are happy that the reviewer noticed the consistency of ab initio and EAM data regarding the viscosity.

' The origin of the seismic attenuation in the inner core has been considered to the presence of partial melting of hcp Fe.'

There are actually quite a few equally fantastic explanations for the attenuation of seismic signal passing through the inner core. However, the common feature of

those explanations is that none of them is plausible.

'Because the viscosity of bcc Fe is similar to that of liquid iron, the authors suggest that the attenuation can be explained the presence of bcc Fe in the inner core.'

It is the only viable explanation for now, considering the thermodynamic stability of the bcc phase under conditions of the inner core. The only low viscosity phase of iron under the inner core conditions is bcc. All other phases are almost ideally elastic. There is also a very recent experimental evidence for the bcc stability in the inner core (Ref. 6 in our paper).

'I feel that the main results of low viscosity of bcc Fe is interesting, and may geophysically important. However, their implication is rather vague and lacking some important arguments.'

The description of implications is now considerably elaborated (in the end of the paper), following the reviewer comments and the editor recommendation.

'I cannot follow the L124-127. Does the author assume the bulk bcc Fe inner core or the presence of bcc Fe inclusion in hcp Fe inner core? I feel the present viscosity value itself cannot eliminate either possibility; therefore, both possibilities should be discussed in the manuscript.'

We do not need inclusions to explain the attenuation – and this is exactly what we say in those lines. The attenuation can be explained by the presence of the bcc iron in the inner core as a bulk. Besides, according to the Gibbs rule, only one phase can be stable at given P and T in the one component system. Impact of impurities (their content is very low) is out of the scope of this paper.

'If the author take the "bulk bcc inner core" scenario, it should be confirmed that the linear relationship between the attenuation and the content of inclusion. Because Singh et al. (2000) assumed up to only 10 % melt fraction, it is clearly inadequate to apply 100 % fraction of bcc Fe.'

We write 'if the relationship is linear' – and it is ideally linear between 0 and 10% percent according to Singh et al. (2000). To satisfy the referee concern we write 'if the linear dependence, observed by Singh et al. (2000) between 0 and 10% is extrapolated to 100 %'. If, for any reason, the linear dependence breaks down, we are still operating within the same range of viscosities as Singh with co-authors.

'Furthermore, even though the authors' previous study predicted the stability of bcc Fe at the Earth's inner core conditions, experimental evidences for the stability of hcp Fe are already reported (Tateno et al., 2010; Sakai et al., 2011). The authors should not ignore these studies.'

We do not ignore these studies. Our recent paper in Nature Geoscience does comment on Tateno et al. 2010 experiments. Other experimentalists have expressed and published concerns that the temperature in their experiments was much lower than reported. Considering that the PT conditions of Sakai et al. 2011 experiments are in a lower range than the reported by Tateno et al. (2010), we do not see why we need also mention Sakai et al. 2011 studies. We also think to tell the story behind Tateno et al and Sakai et al. papers in detail in every our paper would be kind of strange. Certainly, we are not in the position to resolve the arguments between experimentalists, we do believe that they should be left to their own devices.

'On the other hand, if the authors take "bcc Fe inclusion" scenario, both of bcc and hcp phases must be coexist thermodynamically.'

Pure hcp and bcc phases do not coexist at given P and T if one trust thermodynamics (the Gibbs rule).

'I also afraid the difference of viscosity value obtained between AIMD and EAM. I understood that these two values are reasonably agree with each other within the limitation of computational accuracy. Whichever scenario is adopted, the difference may cause the estimated abundance of bcc phase.'

We do not estimate the abundance. We consider bulk bcc Inner Core. However, the referee is right when saying that the difference in AIMD and EAM should be narrowed. We are looking forward when exascale computers and better models for iron will appear. However, at present the data in the paper is the best one can get at the present level of theory development. Besides, AIMD and EAM are consistent in the sense that both methods point to high diffusion and low viscosity. The difference between EAM and AIMD viscosity is negligible in comparison with the correction we make to the existing estimates of the inner core viscosity. Note, that even the experimental estimates span 2 orders of magnitude (10^{20} - 10^{22} Pascal second, Reaman et al., 2012), so our 'computer experiment' is at least as good as the real one.

'Miner comments:

** Units are not unified. In Table 1, the viscosity values are shown in mPa s unit, whereas in main text, they are in kg/m/s.*

** Missing data in Table 1 (BCC, 6000 K, 360 GPa)*

** In Fig.4, there are 4 colors, which may indicate different temperature from 5500 to 7750 K. But I cannot find the correspondence.'*

Units were unified and Pa-s is now used throughout the whole manuscript. The missing data in Table 1 is provided. Fig. 4 caption is modified and now explains the used colors.

Summary of changes.

New simulations are performed. Two new figures (Fig. 1 S and Fig. 2S (a,b,c): Fig. 1 S becomes Fig. 3S) are now provided. Fig 1S illustrates the diffusion in bcc phase and no diffusion in hcp phase for long time in very large samples. Fig. 2S shows the randomness of diffusion in bcc phase (that deals with the concern regarding application of the Stokes-Einstein approach). The implications of the low viscosity of the bcc phase are now explained in detail. 10 new references are added. Typos, units, missing data and other minor things are corrected and/or added. We wish to thank all the reviewers for careful reading of our manuscript and detailed comments that have allowed us to improve the paper.

Reviewers' comments:

Reviewer #2 (Remarks to the Author):

The authors sincerely replied to my concerns. As for the point 2 (Stokes-Einstein vs Kubo-Green formula), if Kubo-Green formula could be used, of course, it would be best. However, the authors pointed out the difficulty to use it due to the slow convergence and added an explanation of limitation of the usage of Stokes-Einstein relation. I think that the explanation is appropriate. I hope that this study would induce an extensive study between Stokes-Einstein and Kubo-Green formula in future. In conclusion, I recommend the editor to publish the present paper.

Reviewer #3 (Remarks to the Author):

In the introduction, the authors mentioned that hcp Fe cannot explain the observed seismic attenuation. Then, they introduced the partial melt model; a small amount (~10%) of melt can explain the attenuation because of its low viscosity. From the present calculation, bcc Fe exhibit a low viscosity, which is comparable to the melt. I think that it means a small amount of bcc Fe can also satisfy the observed attenuation. But the conclusion is different. I have asked the reason why. However, their response seems incomplete and is still lacking the important arguments.

I suspect that their arguments critically depend on unrealistically assumptions such as pure Fe IC and linear viscosity-attenuation relationship. The author did not show any quantitative influence on the attenuation of the IC arising from these assumptions and the difference between AIMD and EAM results, which I previously commented. Therefore, I cannot recommend this article for publication.

Major comments:

The authors stated in the reply as 'There are actually quite a few equally fantastic explanations for the attenuation of seismic signal passing through the inner core. However, the common feature of those explanations is that none of them is plausible'. But if the authors think so, it should be mentioned in the introduction. Basically, the introduction is poor, because of the insufficient review and inappropriate citations. For example, ref. 1 and 2 are referred as 'The Earth solid inner and liquid outer cores consist mainly of iron^{1,2}', but the research target of ref 1 and 2 is attenuation of IC, not the composition and the phase. And they did not appear in the rest of the manuscript. These references should be cited in a proper context. The similar situation is also found in the melting temperature of iron.

The authors suggest in the reply that 'The only low viscosity phase of iron under the inner core conditions is bcc.', but is that really true? Is there any other research that investigates the viscosity of solid iron at high pressure? Is the viscosity of high-P bcc Fe similar to that of low-P bcc Fe? How about the low-P fcc Fe? At least, the authors should compare their results with the literature data of low-P bcc and fcc Fe to emphasize the uniqueness of high-P bcc Fe.

The authors assumed the pure Fe IC, but the IC should contain Ni and light elements. The Gibbs rule suggests that two phases can coexist in the multicomponent system. Furthermore, even assuming a single phase, the presence of impurity should affect the viscosity because the solid viscosity is largely controlled by the slowest diffusion species; the faster the self-diffusion of iron, the more diffusion of other elements becomes important.

Physically unrealistic assumptions, such as linear extrapolation to 100 %, should be eliminated.

In my understandings, the IC structure problem has not solved yet. It is important because present manuscript assume the bcc structure. This is an enough reason to mention the problem. Papers should be completed on their own. I am reading the present manuscript; The authors' previous study does not matter.

Whether it is a computer experiment or a real experiment, the influence of the uncertainty on the geophysical implications must be discussed.

Reply to referees.

Reply to Reviewer #2 comments.

'The authors sincerely replied to my concerns. As for the point 2 (Stokes-Einstein vs Kubo-Green formula), if Kubo-Green formula could be used, of course, it would be best. However, the authors pointed out the difficulty to use it due to the slow convergence and added an explanation of limitation of the usage of Stokes-Einstein relation. I think that the explanation is appropriate. I hope that this study would induce an extensive study between Stokes-Einstein and Kubo-Green formula in future. In conclusion, I recommend the editor to publish the present paper.'

We agree with the notion, that the discovery of the unique almost liquid-like diffusion in solids will ignite their study with different methods, including the Green-Kubo approach. This is very insightful comment. We are also glad to hear that the referee considers our paper to be ready for publication.

Reply to Reviewer #3 comments.(Remarks to the Author):

'In the introduction, the authors mentioned that hcp Fe cannot explain the observed seismic attenuation.'

And we provided the reference where the 'mentioning' is justified.

'Then, they introduced the partial melt model; a small amount (~10%) of melt can explain the attenuation because of its low viscosity.'

No, we did not introduce that model. That is an unfortunate misunderstanding by Reviewer #3. We just mentioned that the paper by Singh et al. explains the attenuation by placing melt inclusions in a solid matrix. Singh et al. introduced the partial melt model, not us. In the manuscript, we provide another reference where the explanation by Singh et al. is shown to be unrealistic because the melt inclusions – if present at some moment of time in the inner core – would be squeezed out of the inner core.

'I think that it means a small amount of bcc Fe can also satisfy the observed attenuation.'

The hcp cannot be the dominant phase of Fe in the inner core – the latest data

points to the stability of the bcc phase.

'But the conclusion is different. I have asked the reason why. However, their response seems incomplete and is still lacking the important arguments.'

We hope that it is clear now and, of course, it is now in the Discussion section of the manuscript. The major reason is stability of bcc under IC conditions.

'I suspect that their arguments critically depend on unrealistically assumptions such as pure Fe IC and linear viscosity-attenuation relationship.'

The inner core is at least 95 percent pure iron (Belonoshko et al., Nat. Geo. 2017). It is quite legitimate to describe the viscosity of the inner core by calculating viscosity of the iron phase thermodynamically stable at the inner core conditions. Not surprisingly, all efforts so far were on measuring/calculating viscosity of pure hcp phase – because the hcp phase was believed to be the stable phase in the inner core. When it appears that the stable phase is bcc, it would be really inconsistent to decide that it is no good to calculate its viscosity. Besides, the amount of light elements within hcp paradigm has to be even larger to match the IC density deficit because the hcp phase is denser than the bcc phase. We also stress that to calculate the viscosity, we do not need to assume the linear viscosity-attenuation relationship. We calculate the viscosity from first principles molecular dynamics and substantiate that by elaborate EAM MD calculations. Nevertheless, in the Stokes approximation and in the range of IC attenuation, the relation between viscosity and attenuation is indeed linear (for some reason the Reviewer #3 calls the Stokes relation physically unrealistic) (Stokes, 1845; see the reference in the manuscript).

'The author did not show any quantitative influence on the attenuation of the IC arising from these assumptions and the difference between AIMD and EAM results'

Again, we did not make any assumptions. We simply noted, that if the linear relation obtained by Singh et al. (Science, 2000) in the interval 0-10% between the viscosity and attenuation holds in the interval 0-100%, the calculated viscosity of iron is a good match to attenuation observed in the IC.

We discussed impact of the difference between EAM and AIMD calculated viscosity. We explicitly said that the difference is about an order of magnitude. Given that we correct the current estimates of the inner core viscosity by 15 to 20 orders of magnitude, such a difference between EAM and AIMD is quite tolerable. We also note, that temperature in the inner core is known with precision of several hundred degrees. Such a temperature difference affects the viscosity by 2 orders of magnitude (Fig. 3). Unless the temperature in the IC is known with a better precision, the effort to calculate viscosity more precisely is pretty much meaningless.

'Major comments:

The authors stated in the reply as 'There are actually quite a few equally fantastic explanations for the attenuation of seismic signal passing through the inner core. However, the common feature of those explanations is that none of them is plausible'. But if the authors think so, it should be mentioned in the introduction.'

It is mentioned in the introduction. The exact wording is somewhat different. We do not use the word 'fantastic', for example, we just write that the 'explanations' contradict to physics (and provide details why).

'Basically, the introduction is poor, because of the insufficient review and inappropriate citations. For example, ref. 1 and 2 are referred as 'The Earth solid inner and liquid outer cores consist mainly of iron1,2', but the research target of ref 1 and 2 is attenuation of IC, not the composition and the phase. And they did not appear in the rest of the manuscript. These references should be cited in a proper context. The similar situation is also found in the melting temperature of iron. '

The reference 1 is to paper by Birch. This is a classical paper on composition of the core, among others subjects. The research target of the ref 1 is not attenuation, contrary to the Reviewer #3 assertion. We suppose the referee might be concerned with references 2 and 3. We agree, that introduction can be improved and partly re-wrote it, including 2 new referenced articles. Concerning the melting temperature of iron, we refer a reader to the assessment made in our recent paper in Nat Geo 2017. Since no major breakthroughs on iron melting has been made since the publication of that article, it is quite legitimate to refer to that assessment instead of copy-pasting that in the present manuscript.

'The authors suggest in the reply that 'The only low viscosity phase of iron under the inner core conditions is bcc.', but is that really true?'

Yes, it is. There is no other solid phase of iron with even remotely similar viscosity.

'Is there any other research that investigates the viscosity of solid iron at high pressure?'

Yes, there is, the Ref. 20 (now 22) is experimental paper and it was cited in the manuscript the Reviewer#3 looked at.

'Is the viscosity of high-P bcc Fe similar to that of low-P bcc Fe?'

No, it is not. The data is in the book by Frost and Ashby, 1982 the reference is now in the manuscript, and the comparison is described in the manuscript.

'How about the low-P fcc Fe?'

Same as above.

'At least, the authors should compare their results with the literature data of low-P bcc and fcc Fe to emphasize the uniqueness of high-P bcc Fe.'

We compare our data on high-PT bcc Fe to the data on Fe at 1 bar. This is a good suggestion and we are thankful to the referee.

'The authors assumed the pure Fe IC, but the IC should contain Ni and light elements. The Gibbs rule suggests that two phases can coexist in the multicomponent system.'

Quantity of light elements does not exceed a first few percent (under 5 percent). Whether Ni is in the IC is unclear, but the generally accepted quantity is about 10%. There is a paper by Dubrovinsky et al (Science), cited now in the manuscript that suggests the stabilization of the bcc structure of Fe on adding 10 percent of Ni. Since the diffusion and low viscosity of Fe is due to particular structure and dynamic stabilization by high temperature, addition of Ni will not change the viscosity estimate. Besides, it is just 10 percent. Of course, even 5 percent of light elements might lead to a complex mineralogical composition of the inner core, however, the major phase is almost pure bcc Fe. Therefore, the impact of Ni and light elements is very likely marginal.

'Furthermore, even assuming a single phase, the presence of impurity should affect the viscosity because the solid viscosity is largely controlled by the slowest diffusion species; the faster the self-diffusion of iron, the more diffusion of other elements becomes important.'

The impact of light elements on viscosity of bcc Fe is best understood if one considers the mechanism of the diffusion in the high-PT bcc Fe. The reason for the diffusion is the dynamic instability of the bcc Fe at high P and low T. The (110) layers sliding one along another in combination with relative disorder caused by a high T leads to motion of atoms along these layers. It is quite different from the mechanism of diffusion in dynamically stable phases where the diffusion is controlled by rare jumps over potential barriers. While in the latter case the referee is quite right, the former mechanism – responsible for the diffusion of Fe in high-PT bcc structure – is not sensitive to light elements as long as impurities do not remove the dynamic instability of the high-P low-T bcc Fe phase. Clearly, under 5 percent of light elements are not capable of that (ab initio MD is routinely performed with 64 atoms – if 3 atoms of these 64 are substituted by carbon that would not change the energy difference between the ideal bcc structure and the wave-mediated (Fig. 1 in Belonoshko et al., Nat Geo 2017)).

'Physically unrealistic assumptions, such as linear extrapolation to 100 %, should be

eliminated.'

First, we do not assume anything to calculate the viscosity. Second, the attenuation is linear on viscosity for low viscosity materials at low attenuations of the IC. In support we cite the paper by Stokes (1845).

'In my understandings, the IC structure problem has not solved yet.'

If the IC structure means the structure of the stable Fe phase under the IC conditions, then the latest data (both theoretical and experimental) points to stability of the bcc Fe. This is written in the manuscript. The manuscript contains References to 5 papers (3 experimental and 2 theoretical) that communicate the stability of bcc iron in the IC. We suppose that is a reasonable number of papers needed to convince the referee that the problem is solved.

'It is important because present manuscript assume the bcc structure.'

Based on the latest data, we consider the bcc Fe phase. It would be really strange if we ignored the latest data in favor of outdated experiments/calculations. Especially when we know which ones and why are simply wrong.

'This is an enough reason to mention the problem.'

We have included the statement where we say that the experimental data on Fe at high-PT is highly controversial. At times, under the same PT conditions and even at the same synchrotron facility (Grenoble) 3 different experimental groups observed 3 different structures of solid Fe (all of those 'discoveries' were published in Nature and Science – apparently, at least two of them are wrong). Our manuscript is not a review on the history of high-PT studies of Fe – though such a review would be very entertaining and illuminating.

'Papers should be completed on their own.'

We could not agree more.

'I am reading the present manuscript; The authors' previous study does not matter. '

We beg to differ! Previous studies do matter! Both ours and published by the referee and any relevant study published by anyone else. Of course they have to be properly cited –if relevant for the subject in the manuscript - and discussed in the manuscript.

'Whether it is a computer experiment or a real experiment, the influence of the

uncertainty on the geophysical implications must be discussed.'

Of course, and we do. See Discussion, please.

Summary of changes.

We have re-written the Introduction to meet the concern of Reviewer#3. In discussion we address the impact of Ni and light elements on the viscosity of the inner core, as well as other possible uncertainties. We show that these uncertainties are highly unlikely to affect our results and geophysical consequences. The suggested by the Reviewer#3 scenario a lot of hcp+a little of bcc is forbidden simply because the stable phase of iron in the core is bcc. We do not see any ground to wave away the stability of bcc. Several references are introduced. The linear attenuation-viscosity relation is substantiated by reference to Stokes paper, even though this is not relevant for viscosity computation in this paper. However, if Reviewer#3 believes it is physically impossible and, therefore, our paper should not be published and the point-by-point response is not sufficient, we provide the reference and explanation anyway.

We wish to thank all the reviewers for careful reading of our manuscript and detailed comments that have allowed us to improve the paper.

Reviewers' comments:

Reviewer #3 (Remarks to the Author):

Previously I commented about unrealistic assumptions: (1) pure Fe inner core and (2) linear extrapolation of the attenuation vs the abundance of low viscosity material. The authors revised the manuscript based on the former, but not on the latter.

The seismic wave attenuation is strongly depends on the geometry of inclusion. Sing et al. (2000) assumed the isolated liquid (low viscosity material) in solid matrix (high viscosity material). But in the present manuscript, the fraction of the low viscosity bcc Fe is 100 %. This is completely different situation, and thus, the linear relationship should be violated. I recommend to reject the paper.

Reply to referees.

Reply to Reviewer #3 comments.(Remarks to the Author):

'Previously I commented about unrealistic assumptions: (1) pure Fe inner core...'

Most of the papers on viscosity of the inner core calculate/measure the viscosity of the pure iron. This for decades was done for hexagonal phase of iron. That is, the viscosity of the inner core was approximated by the viscosity of pure or almost pure hcp iron. However, in light of the latest discoveries it is time to do the same for the bcc. We follow the legitimate approximation, since it is now established that the amount of light elements is below 5 percent. Is it an approximation? Yes, but it is a quite legitimate approximation that is practiced by most of the research community. Fe by most generous estimates constitute more than 90 percent of the material in the inner core. Besides, the referee seems to be happy with our discussion of possible deviation from 100 percent Fe inner core.'

'... and (2) linear extrapolation of the attenuation vs the abundance of low viscosity material.'

The only argument against publication of our paper seems to be the number (2). We actually replied in details to this in our previous report writing that it was a misunderstanding from the referee side. Again, we do not do any, and I repeat – any – assumptions regarding linear or non-linear or exponential or whatever relationship between the attenuation and viscosity. We emphasized that in our previous report (for completeness we included parts of our previous reply to this report – obviously, the referee has chosen to ignore them and now the report qualifies rather as a misleading than a misunderstanding from the referee side). In our paper, the viscosity of Fe in the inner core is computed by first principles molecular dynamics and substantiated by the so-called quasi ab initio molecular dynamics. There is simply no space for the assumption the referee insinuates we made. The only thing we did is that we said that the viscosity of Fe computed by us is similar to the viscosity of the liquid inclusions used by Singh to describe the inner core attenuation (the inclusions were indeed assumed by Singh et al. (2000) to exist in the inner core – but that was the assumption that Singh et al. (2000) made – not us; moreover, the paper by Singh et al. (2000) was shown to be unrealistic – the liquid inclusions would be squeezed out from the inner core – and we provide the reference to that paper in the manuscript). We denounce the Singh et al. (2000) paper (citing another published paper, see manuscript) – therefore, we do not need to do any assumptions regarding the linear extrapolation. It would not be logical from our side to use in support the results

of the paper that was shown to not to provide a physically grounded mechanism for IC attenuation. We believe that the comparison of the viscosity we computed and that of the liquid inclusions in the Singh paper provides a useful information to a reader. However, in the referee eye that qualifies as the reason to reject our paper. This is really hard to understand.

'The authors revised the manuscript based on the former, but not on the latter.'

We did revise our paper both on the former and on the latter. The version we submit is the same the referee inspected. Ironically, the dependence between viscosity and attenuation is indeed linear in the range of the inner core attenuations and we provide mathematically robust arguments in the bottom of the page 7 for that. The referee has chosen not to see it. We believe that it qualifies as the attempt to mislead editors (of course the referee could hardly hope to mislead us). Another possibility is that the referee is not very good in math and could not understand the significance of our arguments.

'The seismic wave attenuation is strongly depends on the geometry of inclusion.'

Correct or not, this is about Singh paper ("Sing", as quoted by referee). It is absolutely irrelevant for our paper, because we do not rely on the existence of inclusions the more so their geometry. We note, however, that Singh et al. assumed a particular geometry to explain the anisotropy rather than the attenuation. To explain the attenuation the geometry is not needed – but again, this is irrelevant.

'Sing et al. (2000) assumed the isolated liquid (low viscosity material) in solid matrix (high viscosity material).'

Again, it is about Singh et al. (2000) paper, not about ours.

'But in the present manuscript, the fraction of the low viscosity bcc Fe is 100 %.'

We have just iron - no solid matrix, no liquid inclusions, no high viscosity material, no low viscosity material. The viscosity is as comes out from very sophisticated first principle MD simulations. Therefore, to describe our approach in the way the referee choses is not very smart at best. Why referee wants us to look like we are describing the core in terms of 100 percent of inclusions (inclusions in what?!) is completely incomprehensible. There are no inclusions at all, rather we are computing an intrinsic physical property that is unique of the Fe bcc polymorph – the stable phase of iron in the Inner Core.

'This is completely different situation, and thus, the linear relationship should be violated.'

We do not care – if the referee believes it is violated, fine (though it is not – see the exercise in math yellow highlighted part at the bottom of page 7), but how does it make our paper incorrect is really mindboggling.

'I recommend to reject the paper.'

First, the referee claims that our paper contains the statements that are not there. Next, the referee insists that those statements are incorrect (ironically, they are correct – the referee seems do not know elementary mathematical analysis – or chooses to not see mathematically robust arguments). Finally, because of those imaginary statements the referee suggests to reject our paper. I have seen my share of not very smart reports, but this one sets my personal best. I do not believe there are such referees – in my opinion it was done on purpose to mislead the Editor and to delay publication.

Summary: The Reviewer #3 either have not understood the paper or intentionally ill-bended its content. Either way, it can be safely disregarded. What he/she criticizes is simply not in the manuscript. To compute the viscosity we do not need to do any assumptions except that inner core is almost pure iron – and it is. The rest comes straightforward (as straightforward as 6 month of running around the clock 500 CPUs can be) from AIMD simulations. If we could calculate the viscosity as the referee suggests we did – it would be much easier for us.

Here we provide the relevant exchange we had with the referee the round before:

'In the introduction, the authors mentioned that hcp Fe cannot explain the observed seismic attenuation.'

And we provided the reference where the 'mentioning' is justified.

'Then, they introduced the partial melt model; a small amount (~10%) of melt can explain the attenuation because of its low viscosity.'

The statement above is just not true. The referee is lying. We thought before that it was misunderstanding on the referee part; now, after our explanation, it qualifies differently.

No, we did not introduce that model. That is an unfortunate misunderstanding by Reviewer #3. We just mentioned that the paper by Singh et al. explains the attenuation by placing melt inclusions in a solid matrix. Singh et al. introduced the partial melt model, not us. In the manuscript, we provide another reference where the explanation by Singh et al. is shown to be unrealistic because the melt

inclusions – if present at some moment of time in the inner core – would be squeezed out of the inner core.

'I think that it means a small amount of bcc Fe can also satisfy the observed attenuation.'

The hcp cannot be the dominant phase of Fe in the inner core – the latest data points to the stability of the bcc phase.

'But the conclusion is different. I have asked the reason why. However, their response seems incomplete and is still lacking the important arguments.'

We hope that it is clear now and, of course, it is now in the Discussion section of the manuscript. The major reason is stability of bcc under IC conditions.

'I suspect that their arguments critically depend on unrealistically assumptions such as pure Fe IC and linear viscosity-attenuation relationship.'

The inner core is at least 95 percent pure iron (Belonoshko et al., Nat. Geo. 2017). It is quite legitimate to describe the viscosity of the inner core by calculating viscosity of the iron phase thermodynamically stable at the inner core conditions. Not surprisingly, all efforts so far were on measuring/calculating viscosity of pure hcp phase – because the hcp phase was believed to be the stable phase in the inner core. When it appears that the stable phase is bcc, it would be really inconsistent to decide that it is no good to calculate its viscosity. Besides, the amount of light elements within hcp paradigm has to be even larger to match the IC density deficit because the hcp phase is denser than the bcc phase. We also stress that to calculate the viscosity, we do not need to assume the linear viscosity-attenuation relationship. We calculate the viscosity from first principles molecular dynamics and substantiate that by elaborate EAM MD calculations. Nevertheless, in the Stokes approximation and in the range of IC attenuation, the relation between viscosity and attenuation is indeed linear (for some reason the Reviewer #3 calls the Stokes relation physically unrealistic) (Stokes, 1845; see the reference in the manuscript).

REVIEWERS' COMMENTS:

Reviewer #2 (Remarks to the Author):

As the authors wrote, the second point raised by Referee 3, “linear extrapolation of the attenuation vs the abundance of low viscosity material” is an explanation that Singh et al. made, not by the present authors. If I have correctly understood what the authors wrote, the authors claim that bcc iron at IC has viscosity low enough to explain attenuation WITHOUT inclusion of low viscosity material.

In fact, in the previous version of the manuscript (158830_1_merged_1530099713.pdf), the paragraph starting at line 132 was somewhat complicated. Discussion of the present authors’ own and Singh et al. seemed to be mixed up. This might have caused Referee 3’s confusion.

In the current version, the paragraph is simplified. I think now the confusion Referee 3 had has been resolved, and the paper can be accepted for publication.

Reply to referees.

Reply to Reviewer #2 comments.(Remarks to the Author):

'As the authors wrote, the second point raised by Referee 3, "linear extrapolation of the attenuation vs the abundance of low viscosity material" is an explanation that Singh et al. made, not by the present authors. If I have correctly understood what the authors wrote, the authors claim that bcc iron at IC has viscosity low enough to explain attenuation WITHOUT inclusion of low viscosity material.

In fact, in the previous version of the manuscript (158830_1_merged_1530099713.pdf), the paragraph starting at line 132 was somewhat complicated. Discussion of the present authors' own and Singh et al. seemed to be mixed up. This might have caused Referee 3's confusion.

In the current version, the paragraph is simplified. I think now the confusion Referee 3 had has been resolved, and the paper can be accepted for publication.'

We appreciate the opinion of the Reviewer #2 and completely agree that in some cases one has to use very easy to understand paragraphs, otherwise it might cause a confusion. It is very satisfying that we managed finally to express ourselves in comprehensible way. We wish to thank all reviewers for their effort.